# Creating a Research-Ready Data Asset version of primary care data for Wales and investigating the impact of COVID-19 on utilisation of primary care services

Hoda Abbasizanjani 📵°*, Stuart Bedston 📵°, Ashley Akbari 📵‡

Population Data Science, Swansea University Medical School, Faculty of Medicine, Health & Life Science, Swansea University, Swansea, United Kingdom

☯ These authors contributed equally to this work.
‡ Senior author
* hoda.abbasizanjani@swansea.ac.uk

## Abstract

### Objectives

We developed an efficient Research-Ready Data Asset (RRDA) for the Welsh Longitudinal General Practice (WLGP) data within the Secure Anonymised Information Linkage Databank to standardise curation, enhance reproducibility, and facilitate research on primary care trends. Using this, we investigated primary care activity trends during and after the COVID-19 pandemic.

### Methods

The RRDA involves cleaning, curation using GP-registration history, and transforming data into a structured, normalised format to support efficient large-scale queries. A comprehensive clinical code look-up was developed, incorporating official, local, and supplementary categories to enhance event classification. To enable patient-practice interaction analysis, a four-layer approach was developed to capture healthcare providers, access mode, interaction type, and event details. We assessed RRDA coverage, defined as the proportion of residents with shared primary care records, stratified by demographic and geographic factors, using longitudinal binomial Generalised Additive Mixed Models (GAMMs). We categorised GP events into key activity types and summarised averaged daily rates per month per 100,000 people (2000–2024), with trends analysed using negative binomial GAMMs.

### Results

Curating 4.6 billion records for 5.1 million people (1990–2024) revealed significant improvements in data quality and completeness over time, with data retention

**Data availability statement:** The data used in this study are available in the SAIL Databank at Swansea University, Swansea, UK. All proposals to use SAIL data are subject to review by an independent Information Governance Review Panel (IGRP). Before any data can be accessed, approval must be given by the IGRP. The IGRP carefully considers each project to ensure the proper and appropriate use of SAIL data. When approved, access is gained through a privacy-protecting trusted research environment (TRE) and remote access system referred to as the SAIL Gateway. SAIL has established an application process to be followed by anyone who would like to access data via SAIL https://saildatabank.com/data/apply-to-work-with-the-data/. This study has been approved by the IGRP as project 0911. The scripts used for data cleaning, curation, and analysis are openly available at https://github.com/SwanseaUniversityDataScience/WLGP_RRDA/.

**Funding:** This work was supported by the ADR Wales programme of work. ADR Wales, part of the ADR UK investment, unites research expertise from Swansea University Medical School and WISERD (Wales Institute of Social and Economic Research and Data) at Cardiff University with analysts from Welsh Government. ADR UK is funded by the Economic and Social Research Council (ESRC), part of UK Research and Innovation. This research was supported by ESRC funding, including Administrative Data Research Wales (ES/W012227/1).

**Competing interests:** The authors have declared that no competing interests exist.

**Abbreviation:** ALF, Anonymised Linkage Field; DHCW, Digital Health and Care Wales; EHR,Electronic health records; GAMM, Generalised Additive Mixed Model; GP, General practice; IGRP, Information Governance Review Panel; ONS, Office for National Statistics; RRDA, Research-Ready Data Asset; SAIL Databank, Secure Anonymised Information Linkage Databank; SNOMED-CT, Systematized Nomenclature of Medicine Clinical Terms; TRE, Trusted Research Environment; WDSD, Welsh Demographic Service Dataset; WIMD, Welsh Index of Multiple Deprivation; WLGP, Welsh Longitudinal General Practice data

increased from 40% to 94%, and patient inclusion from 43% to 98%. Use of SNOMED-CT and local codes increased after Read-V2 discontinuation in 2018, while invalid codes declined—reflecting evolving coding practices and improved data quality. WLGP RRDA coverage rose from 35% in 1990 to 86% in 2024, with regional variation but modest demographic differences. From 2000 to 2024, consultation rates rose by 1.9 times, with post-COVID-19 pandemic levels 8% above 2019. Prescription-only activity doubled with little variation associated with the pandemic. Vaccination rates spiked during the pandemic, and remain 1.8 times above pre-pandemic levels. Other less frequent activities were significantly disrupted during the COVID-19 pandemic but recovered to 2019 levels.

## Conclusions

The WLGP RRDA improves the usability of primary care data, supporting timely, scalable analysis of healthcare delivery and system-level trends.

## Introduction

Primary care is the foundation of healthcare delivery, providing continuous, comprehensive, and accessible services for a broad range of health concerns. General practices (GPs) serve as the first point of contact for most patients, managing acute and chronic conditions, coordinating specialist referrals, and delivering preventive care [1]. The efficiency and accessibility of primary care services are essential for maintaining population health and reducing the burden on secondary and emergency care services [2,3]. The widespread adoption of electronic health records (EHRs) has transformed how primary care is documented, enabling more structured recording of patient-provider interactions. EHRs contain valuable information on consultations, diagnoses, prescriptions, test results, and referrals. These routinely collected data serve as a valuable resource for research, public health monitoring, and healthcare planning, offering insights into service utilisation, trends in disease management, and healthcare disparities [4,5].

The COVID-19 pandemic profoundly disrupted primary care delivery, fundamentally altering two core dimensions of how patients interact with general practice: the **mode of access** (e.g., face-to-face vs remote) and the **type of activity** (e.g., consultations, prescribing, vaccinations). These shifts were shaped both by public health responses to reduce viral transmission and by broader structural and technological adaptations. In Wales, this transition was facilitated by digital health infrastructure developed prior to the pandemic, including the Informed Health and Care strategy: A Digital Health and Social Care Strategy for Wales (2015), which aimed to embed digital tools across the NHS [6]. In March 2020, a system enabling all GPs in Wales to offer online consultations was rolled out nationally, first trialled in the Aneurin Bevan University Health Board [7].

Changes to **access mode** were immediate and widespread. The majority of consultations shifted from in-person to remote format, primarily telephone-based.

While this preserved access during lockdowns, it also raised concerns about equity, diagnostic accuracy, and continuity of care. Several studies have examined these access-related changes. A large-scale OpenSAFELY study showed remote consultations increased during the pandemic and were more likely among women, younger adults, and individuals from more deprived and White ethnic backgrounds [8]. Efforts to support digital access had inconsistent effectiveness across practices [9], and remote care disproportionately affected older adults and patients with complex needs [10]. Migrants in England were already less likely to use primary care before COVID-19, and the pandemic worsened these disparities, underscoring the importance of culturally and digitally inclusive models [11].

The pandemic also led to substantial changes in the types of activities delivered in primary care. Reductions in routine care and long-term condition management were widely reported. In Wales, a sharp decline in the incidence of 17 long-term conditions, such as diabetes, chronic obstructive pulmonary disease, hypertension, and depression, has been shown, suggesting a large backlog of undiagnosed patients [12]. A large-scale study of community-dispensed cardiovascular disease medications from England, Scotland and Wales found a substantial number of individuals likely missed treatment for major cardiovascular disease risk factors, with only partial recovery in medication initiation post-pandemic [13]. These disruptions were not confined to diagnoses: preventive services, routine monitoring, and prescribing practices also changed. Patients with rheumatoid arthritis experienced reduced contact frequency and variable monitoring across the UK [14]. In Wales, there was a marked reduction in antibiotic dispensing during the early pandemic months, likely due to both reduced infection transmission and lower in-person consultation rates [15]. Community medication dispensing in Wales showed disruptions in routine prescribing during peak pandemic periods [16]. Across the UK, the COVID-19 pandemic disrupted cardiovascular disease prevention and management, with a large drop in new prescriptions for antihypertensives and lipid-lowering medications likely contributing to future excess cardiovascular events [13]. In England, prescription volumes for chronic conditions fell during the first pandemic year, with only partial recovery by 2021 [17].

National statistics reflect these shifts in clinical activity and access mode. Comparing GP appointments in November from 2019 to 2024, patient interaction patterns with GPs in England shifted significantly, reflecting lasting changes brought about by the COVID-19 pandemic. While total appointment volumes (including vaccinations) increased from 26.4 million to 31.4 million. Face-to-face consultations, which had dropped sharply during the COVID-19 pandemic (from 21.5 million in November 2019 to 13.8 million in November 2020), recovered partially and stabilised around 20.7 million by 2024, still slightly below pre-pandemic levels. Telephone consultations more than doubled during the COVID-19 pandemic, peaking near 10 million, and although they declined slightly in later years, they remained at 7.67 million in November 2024, well above 2019 levels. Home visits, after dipping in 2020, rebounded steadily from 0.3 million in November 2019 to 0.4 million by November 2024. Most notably, "Online/video" appointments rose nearly tenfold—from just 0.18 million in November 2019 to 1.7 million in November 2024, signalling a growing reliance on digital-first services in primary care [18]. These trends indicate that while face-to-face care remains the foundation of general practice, a hybrid model has become firmly embedded in post-pandemic NHS service delivery.

In Wales, 64.7% of GP appointments from April 2023 to March 2024 in Wales were conducted face-to-face, while 34.0% were delivered remotely, marking a partial return to in-person care from COVID-19 pandemic lows [19]. The report also found that chronic and planned care more commonly occurred face-to-face, while remote consultations were more evenly split for chronic, planned or non-acute reasons (50.6%) and for urgent or acute reasons (49.4%).

Despite these insights, there remain critical gaps in understanding the scale and nature of change in primary care activity in the UK. Much of the existing research has focused on specific conditions, patient groups, or consultation types. Less is known about broader trends in core primary care functions, such as consultations, prescribing, and vaccination, across the Welsh population. Additionally, inconsistent coding of access mode or interaction type complicates efforts to track care delivery over time [20,21].

We aim to investigate changes in primary care activities and patient-practice interactions (e.g., consultations, prescriptions) in Wales during and after the COVID-19 pandemic, compared to pre-pandemic trends, using primary care data

available within the Secure Anonymised Information Linkage (SAIL) Databank, the national trusted research environment (TRE) for Wales. Understanding these patterns can offer valuable insights into patient access to primary care, the effectiveness of different modes of delivery, and the overall performance of the healthcare system during a period of significant disruption. Investigating whether these patterns have returned to pre-pandemic levels or reflect lasting changes is essential for assessing the long-term impact of the pandemic on healthcare delivery and patient behaviour.

As part of this study, we created and analysed a nationwide, standardised Research-Ready Data Asset (RRDA) for the Welsh Longitudinal General Practice (WLGP) data within the SAIL Databank. The motivation behind this is twofold. Firstly, we aimed to streamline the processing time, complexity, duplication and efficiency for use in research by eliminating the time-consuming task of data curation for primary care data, thereby enabling new studies to start more quickly. Secondly, by standardising the curation process and creating a shared data asset, we can ensure that research findings are more easily comparable and reproducible. Using this RRDA, we then assessed population coverage by analysing data availability annually from 1990 to 2024, and analysed longitudinal trends of patient-practice activity.

## Methods

### Ethics approval and consent to participate

Approval for the use of anonymised data in this study, provisioned within the SAIL Databank, was granted by an independent Information Governance Review Panel (IGRP) under project 0911. The IGRP has a membership comprised of senior representatives from the British Medical Association, the National Research Ethics Service, Public Health Wales and Digital Health and Care Wales (DHCW). The usage of additional data was granted by each respective data owner. The SAIL Databank is compliant with General Data Protection Regulations and the UK Data Protection Act.

### WLGP RRDA development

We used anonymised individual-level linked data within the SAIL Databank to develop an efficient RRDA version for the WLGP data. The WLGP includes records for all patients registered with Welsh GPs, for the GPs who have agreed to share data with the SAIL Databank, as GPs are the data owners and must individually consent to contribute [22]. Furthermore, individual patients can opt out of having their anonymised records included in SAIL by making a request to their GP [23]. Currently, WLGP covers 86% of the Welsh population and 83% of GPs in Wales [24]. Each record includes the unique encrypted person identifier in SAIL (known as Anonymised Linkage Field (ALF)), event date, anonymised practice identifier (practice ID), event clinical code, and clinical value (such as blood pressure reading, or a lab test result where applicable), along with additional relevant information [24].

The time span of data available from each practice varies, depending on when electronic record-keeping began and how recently data were submitted. Additionally, patient registration history can be complex, with individuals potentially having multiple registrations with different GPs and moving in and out of Wales over time [22]. This variability in data collection and patient movement presents challenges in data curation and consistency.

Although the WLGP data provides a rich source of primary care information, several potential data quality issues can arise. These include duplicates, missing or invalid entries due to data entry errors, and incomplete records. Some records may also relate to temporary patients or individuals not permanently registered with a practice in Wales. Furthermore, GP-to-GP transfer processes may result in records being reinserted as duplicates or assigned to new practices, complicating data curation efforts [25].

The WLGP data is structured in a long-format event-list format, which currently contains over 4.6 billion records. This structure poses challenges when conducting large-scale queries, as each record is processed individually, leading to inefficiencies.

Most clinical events in the WLGP are recorded using the official Read V2 coding system at the 5-character level. However, a small proportion of events are recorded using 7-character Read codes, Systematized Nomenclature of Medicine

Clinical Terms (SNOMED-CT), or 'local codes'. Primary care software systems use local codes for specific purposes; however, as they are not official codes, they are not listed or published in official code browsers. These local codes became especially relevant after the discontinuation of Read codes in 2018. For example, new clinical concepts such as COVID-19 and Long COVID are now identified using SNOMED codes, which are not yet available in Welsh primary or secondary care EHR systems. Instead, these are captured as local codes in individual software systems [26].

Currently in Wales, two primary care software providers, EMIS and Vision, are used to record details of consultations and other activities [27,28]. These details are securely acquired into the SAIL Databank via the standardised data acquisition, anonymisation, and approval processes. In addition to the local codes from EMIS and Vision, the WLGP may also contain unverifiable codes, which originate from previous software systems used by some practices in Wales, which lack descriptions, posing further challenges for data standardisation.

To address these challenges and enhance the usability of the data, we created an RRDA for the entire WLGP data source and all available primary care data since 1990, covering the population of Wales. The development of the RRDA involves six main steps (see https://github.com/SwanseaUniversityDataScience/WLGP_RRDA/ for the scripts):

### Step 1 (Data cleaning)

Records (i.e., individual event entries in the WLGP data) with missing ALF, event date, or practice ID were removed, as these are essential for identifying and linking patient data. The WLGP data does not include records with missing event codes.

### Step 2 (GP registration-based validation and extraction of correct practice ID)

The Welsh Demographic Service Dataset (WDSD) provides a history of individuals' GP registrations across all of Wales, this includes all practices in Wales longitudinally, not just the practices contributing data to SAIL, as WDSD is centrally managed by DHCW, formerly known as the NHS Wales Informatics Service. This data source includes multiple records for individuals, as they may move in and out of Wales, relocate within Wales, or register with different practices while residing at the same address [22,29].

We excluded events for individuals who did not have a valid GP registration in Wales at the time the event occurred. Additionally, the WDSD was used to determine the correct practice ID at the time of each event in the WLGP data, allowing us to update practice IDs when necessary, such as in cases of GP-to-GP transferred records.

### Step 3 (Removal of exact duplicates)

We eliminated event records where all fields extracted from the original WLGP were identical, ensuring the data only retained unique entries.

### Step 4 (De-duplication of GP-to-GP transferred records)

We identified GP-to-GP transferred records, event records identical except for the practice ID, by comparing practice IDs from WLGP and WDSD at the time of the event, and retained the appropriate version.

### Step 5 (Creation of a comprehensive look-up of primary care clinical codes)

In collaboration with the SAIL Databank and DHCW, we confirmed with the current GP software system providers for Wales (EMIS and Vision) a list of their respective local codes as of 2023. Additionally, we identified a list of further EMIS clinical codes from the UK Biobank [30,31] as well as EMIS prescription codes from the same source [32]. Additional Vision and Read V2 codes were obtained from some external sources [33,34].

We then created a comprehensive look-up table that includes all official Read V2 and SNOMED codes (available within the SAIL Databank), as well as local EMIS and Vision codes provided by DHCW or identified from external

sources. The look-up contains clinical codes, their code type, descriptions, source and any additional categorisation available from the external sources. In addition to the recognised code types, we included two supplementary categories in the clinical code look-up: **'Blank or invalid codes'**, referring codes that were blank or failed basic format validation; and **'Unknown codes'**, which appeared structurally valid but lacked known descriptions or classification, often originating from legacy or undocumented local codes used by software providers, or codes set up by practices themselves. The inclusion of these supplementary categories allowed us to classify all clinical events in the RRDA and enhanced the interpretability of this data.

Some challenges arose during the creation of the look-up table. For instance, we encountered cases where the same SNOMED-CT and Vision local codes had different descriptions. To address this, we standardised the format of Vision codes to the 5-character level, either by removing extra characters or adding a "." to the right if the code was shorter than five characters. A similar approach was applied to EMIS and Vision codes. We also found overlap between Read V2 and EMIS codes, which we resolved by treating the overlapping codes as Read codes and removing them from the EMIS list. In general, to manage duplicate codes across different code types —whether arising from truncation or padding, or from overlapping entries across sources (e.g., EMIS codes appearing both in DHCW and UK Biobank lists)— we prioritised the codes based on their type, retaining the first code type encountered in the following order: official Read V2 codes, official SNOMED codes, DHCW local Vision codes, DHCW local EMIS codes, additional Read or Vision codes, and additional EMIS codes from the UK Biobank. See S1 File in the Supporting information for some examples.

**Step 6 (Normalising WLGP RRDA)**

In order to reduce data redundancy and improve the efficiency of large-scale queries, we normalised the WLGP RRDA by restructuring it into a three-table format. A record in the **person-day** table was defined as a unique combination of an ALF, event date, and practice ID.

The normalising process involved creating unique integer-based keys for **clinical codes** and **person-day events**, which improved indexing, reduced storage requirements, and enhanced query performance. The structure consists of three linked tables:

1. The **person-day table**, which contains the unique event identifier (event ID), ALF (enabling linkage to other data sources within SAIL), event date, and practice ID. Each record in this table represents a unique person-day interaction with primary care.

2. The **event table**, which stores all recorded clinical events associated with a given event ID. Each record in this table includes the unique clinical code identifier (clinical code ID) and associated clinical values (e.g., test results or numerical readings, where applicable).

3. The **clinical code look-up table**, which maps each clinical code ID to its corresponding description, coding system (e.g., Read V2, SNOMED-CT, or local codes), and any additional categorisation.

By structuring the data in this way, each person-day event can be efficiently linked to multiple clinical events, while clinical codes are stored separately, reducing redundancy and improving scalability. This design allows for more flexible and faster querying of primary care records while maintaining the integrity of the data (Fig 1).

**Multi-layer approach for identifying types of patient-practice interactions**

The WLGP data includes coded clinical events encompassing diagnoses, medical history, symptoms, lab results, procedures, prescriptions issued by general practitioners, referrals, and a range of administrative codes (e.g., patient registrations and demographic information). However, the complexity of primary care activities necessitates a structured approach to accurately classify patient-practice engagements within the WLGP data.

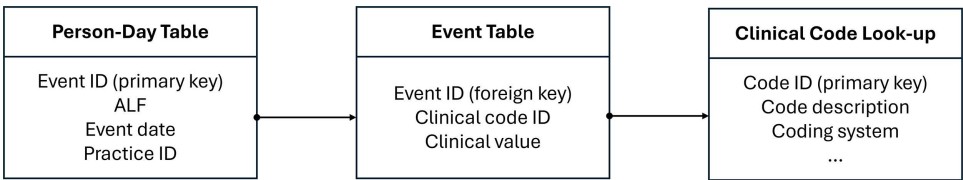

**Fig 1. Structure of normalised WLGP RRDA tables.** A primary key uniquely identifies each row in a table, while a foreign key is a column in one table that links to the primary key in another table.

To systematically capture the complexity of patient interactions, we developed a four-layer classification system that assigns healthcare provider type, access mode, interaction type, and specific interaction details to individual clinical codes (Table 1). This approach enables robust patient-practice interaction analysis by accounting for the various ways in which healthcare interactions are documented.

The **Care Provider Layer** identifies the type of care provider responsible for the recorded interaction. While most records in WLGP relate to an interaction within primary care settings, some may document interactions with other care providers, such as secondary care or community services. The **Access Mode Layer,** defined only for primary care interactions, classifies how the patient accessed care. It sets the overall context for the interaction and includes the following categories: face-to-face (e.g., in-practice visits, immunisation or vaccination, and home visits), remote (e.g., telephone or video appointments), clinical data with unmatched access mode (patient's clinical records where the access type cannot be explicitly determined), admin data (e.g., registration updates, or demographic entries), and currently cannot be assigned (non-clinical records with no available information on interaction type). The **Interaction Type Layer** provides a more specific classification within each access mode, detailing the method or setting of the interaction (Table 1). The **Interaction Details Layer** offers a granular breakdown of specific activities or procedures performed within certain interaction types (Table 1). This ensures that significant clinical activities are accurately represented, enhancing the granularity of primary care interaction data.

To implement this system, we leveraged hierarchies and categories of both official and local clinical codes (where available), along with free-text search, to ensure accurate classification while maintaining flexibility for evolving clinical documentation practices. For this part of the work, we included only official Read V2 codes, local EMIS and Vision codes, additional Read or Vision codes, and supplementary EMIS codes from the UK Biobank. SNOMED codes were excluded from this classification step as they account for only 0.1% of WLGP RRDA records and use a different hierarchical structure from Read codes, with over one million codes, making their inclusion disproportionately complex for limited benefit. Overall, this classification approach enhances the interpretability of primary care records and provides a robust framework for analysing patient-provider interactions at scale.

Using the multi-layer approach, we first assigned the care provider, access mode, interaction type, and interaction details to each clinical event. This classification enabled us to categorise daily patient interactions (i.e., person-day events) into key activity types, considering only those that included at least one interaction with a primary care provider. While our framework includes classification by access mode (e.g., face-to-face, remote) and interaction type (e.g., practice visit, home visit, telephone call), the WLGP data contains limited information on how patients interacted with primary care. Although some Read V2 and local codes exist to capture access mode and interaction type, these were often sparsely and inconsistently recorded. As a result, for a large proportion of events, these details could not be reliably determined. To ensure broader coverage and more consistent categorisation, we focused on activity types (such as consultations, vaccinations, patient review or monitoring) which can be more robustly inferred from the available clinical content. A person-day event was classified as a **consultation** if it met one of the following criteria:

**Table 1. Hierarchical classification of patient-practice interactions.**

| Layer 1:<br>Care Provider | Layer 2:<br>Access Mode | Layer 3:<br>Interaction Type | Layer 4:<br>Interaction Details |
|---|---|---|---|
| Primary care | Face-to-face | In-practice visit | Examination or sign |
| | | | Observation |
| | | | Screening or assessment |
| | | | Laboratory procedure |
| | | | Therapeutic procedure |
| | | | Other face-to-face interactions within practice setting |
| | | Immunisation/vaccination | NA |
| | | Dental service | NA |
| | | Pharmacy | NA |
| | | Home visit | NA |
| | Remote | Phone call with patient | NA |
| | | Text message, email, or letter (to/from patient) | NA |
| | | Other remote interactions | NA |
| | Clinical data with unmatched access mode | Clinical activities | Drug therapy or prescription |
| | | | Lab test request or result |
| | | | Chronic disease monitoring |
| | | | Patient monitoring |
| | | | Maternal or child health |
| | | | Diagnosis |
| | | | History or symptom |
| | | | Referral |
| | | | Counselling or health education/ promotion |
| | | | Patient review or primary prevention |
| | | | Other clinical documentation |
| | Admin related data | Patient admin data | Patient sociodemographic or registration data |
| | | | Certificate |
| | | | Failed encounter |
| | | | Other patient admin data |
| | | Other admin data | NA |
| | Cannot be assigned | NA | NA |
| Secondary care | NA | NA | NA |
| Community care | NA | NA | NA |
| Cannot be assigned | NA | NA | NA |

- It included at least one record related to an examination, signs or symptoms, observations, history of disease, diagnosis, lab procedure, lab test request or result, screening or assessment, chronic disease monitoring, patient monitoring or review, therapeutic procedure, certificate, maternal or child-related records, counselling and health promotion, or referral.

- It included a drug therapy or prescription related record with at least one record with an access mode recorded remote, or with an interaction type classified as in-practice or home visit.

Person-day events that exclusively contained drug therapy or prescription records were classified as **prescription only**. **Vaccinations** were defined as primary care person-day events containing a vaccination record, provided there was no record of a pharmacy visit, dental service access, or failed encounter. Additionally, primary care person-day events containing only administrative records were categorised as **administrative** person-day events. We also defined additional categories of primary care person-day events as follows:

- **Certificate issuance**: Events containing any record related to the issuance of a certificate or fit-note.

- **Patient review or monitoring**: Events containing records related to chronic disease monitoring, patient monitoring, or review.

- **Screening and assessment**: Events containing records related to patient screening or assessment.

- **Failed encounters**: Events exclusively containing a record of a failed primary care encounter, defined as instances where a patient did not attend or was not brought to a scheduled appointment, where an appointment was cancelled by either the patient or the healthcare provider, where the patient could not be contacted, or where an invitation or consultation was declined by the patient.

These categories are not necessarily mutually exclusive; a person-day event may belong to multiple categories if its records meet the criteria for more than one (for example, both "Patient review or monitoring" and "Certificate issuance").

### Analysis of WLGP population coverage trends

We analysed GP registration coverage from 1990 to 2024. For each year, we calculated the number of individuals living in Wales and the number with WLGP coverage, stratified by sex, age, Welsh Index of Multiple Deprivation (WIMD, version 2019) quintile, and health board. We defined someone as living in Wales if they had a Welsh Lower layer Super Output Areas on 1st July, and had been continuously living in Wales for at least three months prior or were born in the last three months. WLGP coverage was defined as the number of individuals living in Wales whose GP records were available within the SAIL Databank, either because their current GP at the time was actively sharing data with SAIL, or because historical records were later added through data shared by a future GP participating with SAIL.

To estimate associations between WLGP coverage and demographic factors (sex, age, WIMD quintile, and health board), we fitted longitudinal binomial Generalised Additive Mixed Models (GAMMs). This approach incorporated a thin-plate regression spline to capture the general trend over the years, and an autoregressive covariance structure of order 1, AR(1), between year-on-year observations nested within each level of the covariates to capture the dependency between repeat observations across the characteristics. We estimated coefficients and 95% confidence intervals for each of the covariates, both unadjusted and adjusted for the other factors stated. S1–S3 Tables and S1–S3 Figs in the Supporting information contain an expanded set of descriptive counts of all those living in Wales, as well as those registered with a Welsh GP, and stratify description of GP coverage over time by sex, age and WIMD.

### Analysis of GP activity trends

We conducted a retrospective cohort study of all Welsh residents attending Welsh general practices with linked records to analyse national trends in patient-practice activity between January 2000 to December 2024. To identify the study population, we used residence records to identify all those living in Wales and our RRDA to identify those registered with a Welsh primary care GP at any time during the study period.

The unit of analysis was one calendar month, with the outcome being the average daily number of events that month per 100,000 people for a given activity. We categorised each person-day of GP activity into the following eight groups: Administrative only; Certificates; Consultations; Failed encounters; Prescription only events; Patient review, monitoring

and chronic disease monitoring; Screening and assessments; and Vaccinations. We counted the number of person-days each month for each category and examined trends by fitting separate negative binomial GAMMs. Model specification included a log of the underlying GP-registered population as an offset term, a cyclic penalised cubic regression spline for the 12-month seasonal trend, a thin-plate regression spline for the general trend, and an AR(1) covariance structure between month-on-month observations. We omitted observations from 2020 and 2021 from model fitting due to COVID-19 pandemic restrictions and disruptions. For vaccinations, we excluded 2021 and 2022 instead, as this period coincided with the rollout of the UK-wide COVID-19 vaccination programme. This caused a huge increase in vaccination activity over the two-year period, as population subgroups became eligible for the initial two-dose schedule, as well as booster doses.

To estimate the extent to which activity rates changed after the COVID-19 pandemic compared to before, another series of GAMMs was fitted with the same specification but with year as a categorical covariate (with 2019 as the reference level), thus providing us with rate ratios for all other years.

S4-S19 Figs in the Supporting information contain results from the initial exploratory analysis of the GP activity trends consisting of: STL decomposition of the rates, Autocorrelation Function plots, as well as plots of the fitted spline functions for calendar month and the general trend from the GAMM results.

## Results

### RRDA vs original WLGP

The WLGP contains more than 4.6 billion clinical event records for approximately 5.1 million individuals from 1990 to 2024. Of those, 98.3% of records are linkable to individuals with valid GP registration records in WDSD. Following compressing records and de-duplicating GP-to-GP transferred records, 86.7% of clinical event records were retained in the RRDA (Fig 2).

Substantial improvements in data quality and population coverage were observed over time in the underlying WLGP data, as revealed through the RRDA curation process. The data retention rate, i.e., the proportion of records retained after cleaning and validation, increased markedly from 40.2% in the early 1990s to 93.8% by 2024. Similarly, the proportion of patients retained in the RRDA increased from 43.2% to a peak of 98.1%, reflecting improved accuracy and linkage reliability over time. A slight decline was noted in the final year of the study in the patient-level retention proportion, the cause of which remains unclear and may relate to delays in data flow or recent registration changes. Overall, these trends indicate a consistent enhancement in data quality and the representativeness of the Welsh population (Fig 3).

Normalising the WLGP data as part of the RRDA development substantially improved query performance and analytical efficiency. The three-table relational format facilitated more scalable and iterative analysis compared to the raw format, particularly when working with longitudinal large time spans or full-population extracts.

### Comprehensive clinical code look-up

Table 2 and Fig 4 (A, B) show an overview of the clinical code types incorporated into the WLGP RRDA clinical code look-up and their usage per year, expressed as the proportion of distinct codes and records. While Read V2 is the official coding system for the WLGP, we found that up to 7.2% of records per year used an alternative coding system, such as SNOMED-CT or local EMIS/Vision codes, with a modest increase to around 11.1% observed in 2021 (Fig 4B). This increase largely reflects changes in coding practice following the official discontinuation of Read codes in 2018. Newer concepts, such as those related to COVID-19, were introduced in SNOMED-CT, but as SNOMED-CT is not yet fully integrated into Welsh EHR systems, they are often recorded using local codes. Over time, there was a clear increase in the use of SNOMED-CT and local EMIS/Vision codes, particularly from 2020 onwards. This shift was accompanied by a decline in the proportion of 'blank or invalid' codes during the 1990s and early 2000s (Fig 4B) and standardisation across Welsh general practices.

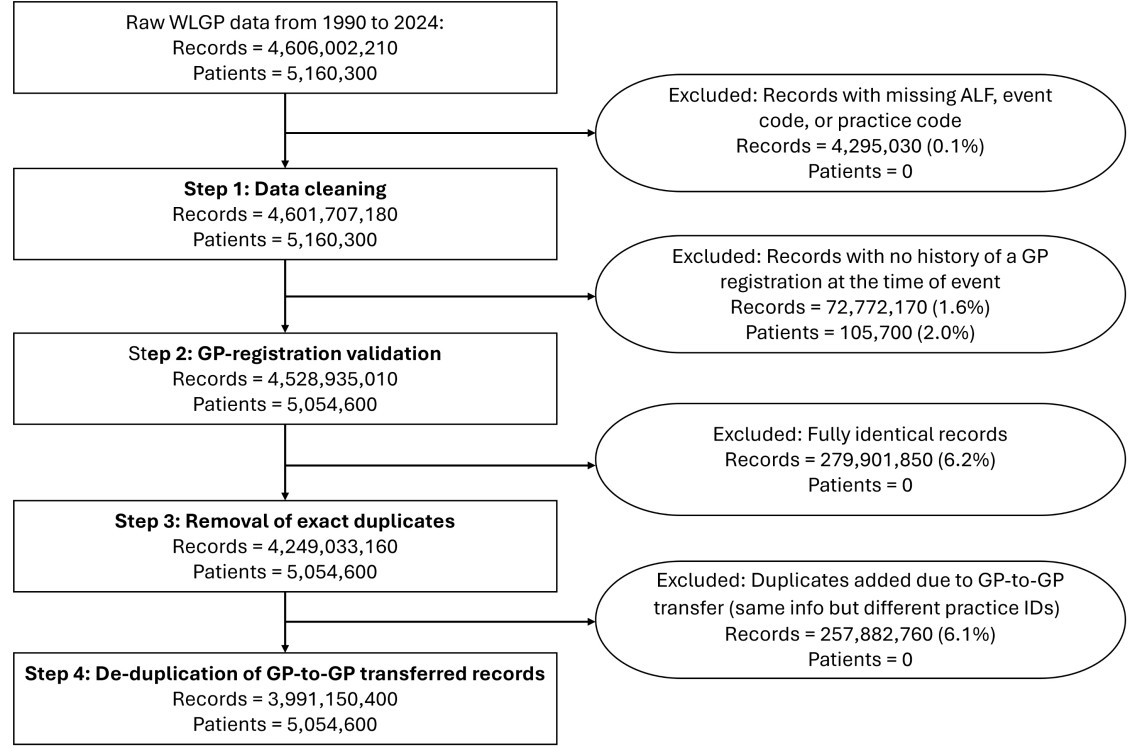

**Fig 2. Consort diagram showing data cleaning and curation steps in the creation of the WLGP RRDA (all counts <10 have been masked, and other counts have been rounded to the nearest 10).**

## GP registration coverage in WLGP RRDA

Over the whole period 1990–2024, we have 5.9 million individuals recorded as living or lived in Wales as part of the annual snapshot population each year, with 4,921,630 (86.8%) having linked primary care records in the WLGP RRDA due to being registered at a SAIL data-sharing GP at some point within that period. However, residential coverage and primary care linkage were not uniform over that period (Fig 5), with the population coverage initially being below the Office for National Statistics (ONS) mid-year population estimate and exceeding it for the first time in 1994 by 9,790. From then, population coverage has mapped closely with the ONS estimates, albeit always slightly above, approximately +1.9%. Of those living in Wales in 1990, we found only 35.2% had linked primary care records (Table 3). This then greatly increased to 85.1% by 2000, from there it remained relatively stable at 86.2% in 2024. For a more granular longitudinal breakdown, see S1-S3 Tables in the Supporting information.

Linkage availability was found to vary most by health board over time, in particular, between 1990 and 1999, which also benefited the most from records being made available via the sharing of historic records (Table 3, Fig 5). However, by 2024, three of the seven health boards had record linkage for at least 90% of residents (Cardiff & Vale, Swansea Bay and Cwm Taf Morgannwg), three between 82% and 84% (Hywel Dda, Betsi Cadwaladr, Aneurin Bevan), and Powys having the least coverage (44%) but also the smallest population (135,070 people, compared to Betsi Cadwaladr with 705,900 people). This ranking of health board coverage persisted after adjusting for sex, age and WIMD, where we estimated adjusted odds ratios for Powys to be 0.18 (95% CI 0.180–0.182) times that of Betsi Cadwaladr, 2.35 (95% CI 2.34–2.36) for Swansea Bay and 5.31 (95% CI 5.28–5.34) for Cwm Taf Morgannwg Health Board.

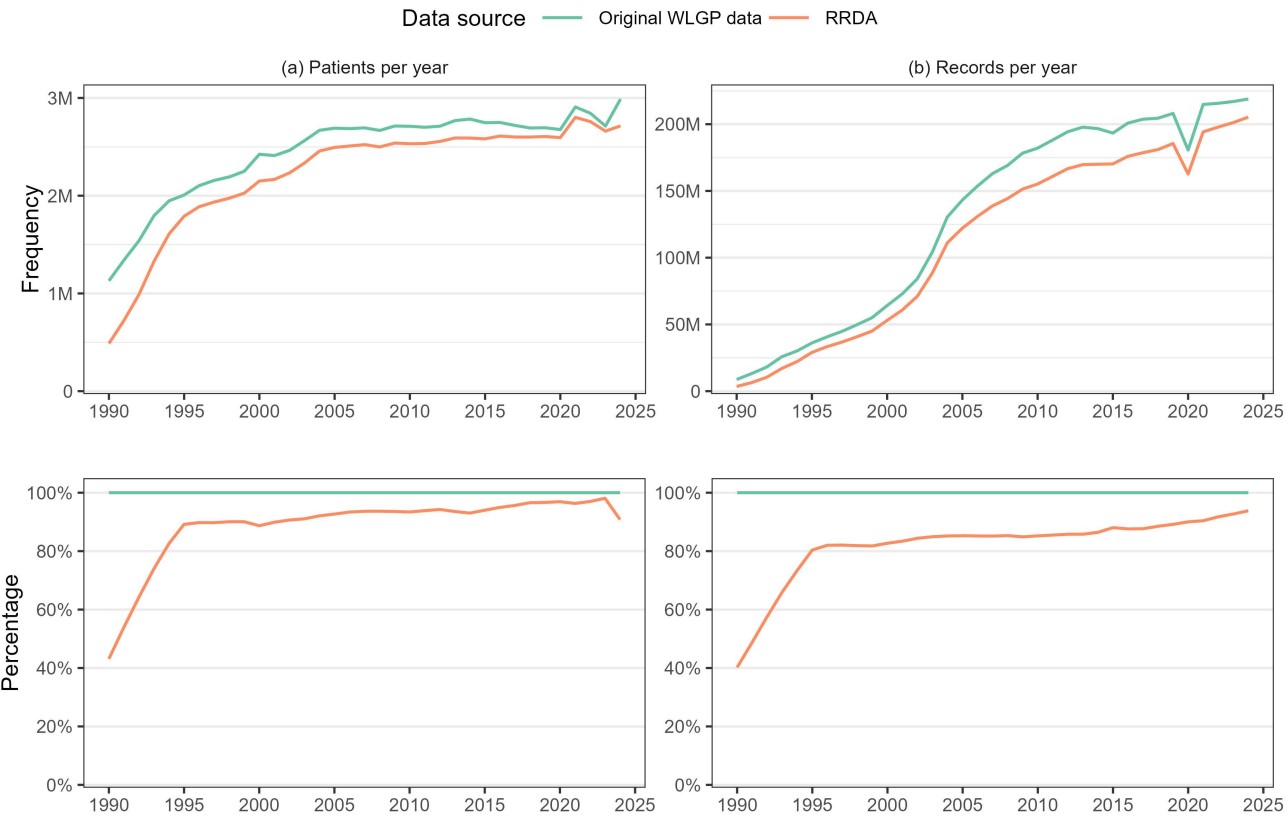

**Fig 3. Frequency and % of patients and clinical records in the original WLGP and its RRDA version over time.**

**Table 2. Summary of clinical code types added to the WLGP RRDA look-up (excluding official Read V2 codes).**

| Code type | Source | Distinct codes added to the RRDA look-up | Records mapped in WLGP RRDA (1990–2024) [a] |
|---|---|---|---|
| SNOMED-CT | SAIL | 1,124,798 | 5,096,230 (0.1%) |
| Local Vision codes | DHCW | 982 | 7,227,230 (0.2%) |
| Local EMIS codes | DHCW | 51,783 | 82,789,970 (2.1%) |
| Additional Read or Vision codes | [33,34] | 96,752 | 10,864,850 (0.3%) |
| Additional EMIS | UK Biobank [30–32] | 19,216 | 959,990 (0.0%) |
| Blank or invalid codes | – | 90 | 18,376,200 (0.5%) |
| Unknown type | – | 397,657 | 54,927,410 (1.4%) |
| **Total** | – | **1,691,278** | **180,241,880 (4.5%)** |

[a]Rounded to nearest 10

Proportionally, linkage availability was very similar between males and females, as well as across the age groups, and by residential WIMD quintiles (Fig 6). Females were found to have a slightly higher coverage than that of males (aOR 1.115, 95% CI 1.112–1.118). Those aged 0−15 years had the highest proportion of coverage, with those aged 16−34 and 35−49 having a slightly lower proportion (aOR 0.929 and 0.914), and 50−64 having comparably lower coverage (aOR 0.823, 95% CI 0.825–0.832), and those aged 65−110 years old having relatively the worst coverage (aOR 0.653 95% CI

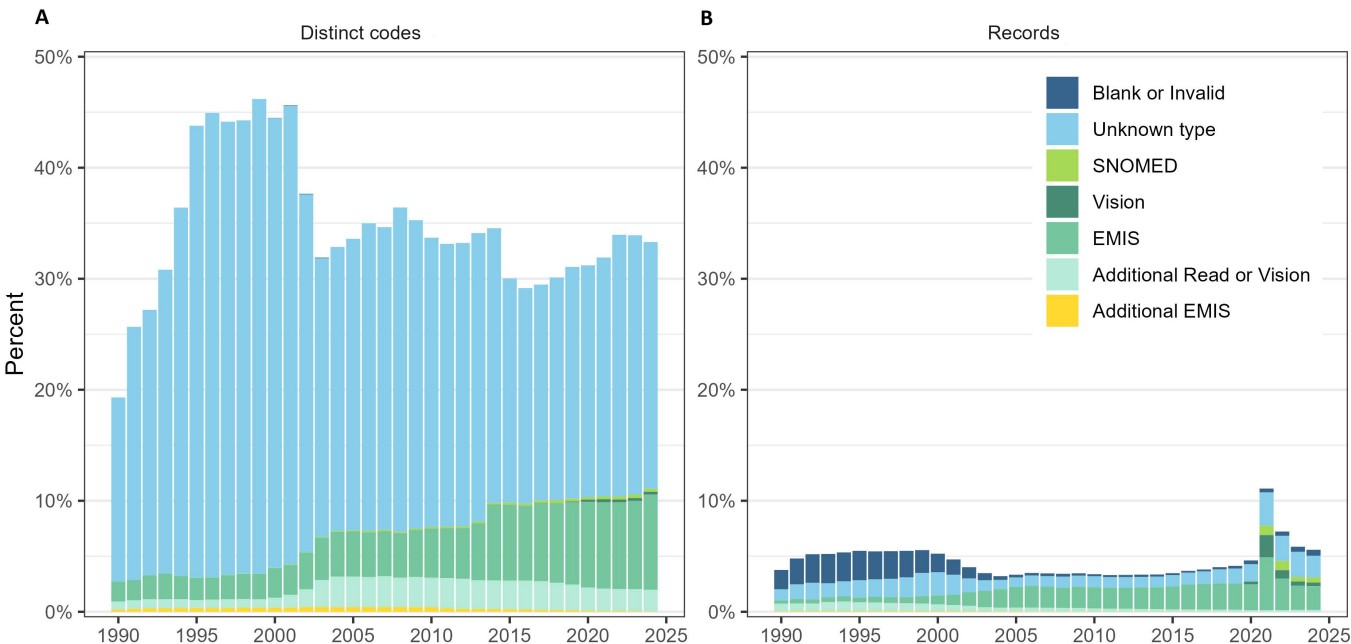

**Fig 4. Usage of clinical code types other than official Read V2 codes by year in the WLGP RRDA. (A)** % of distinct clinical codes. **(B)** % of records mapped to clinical codes types. Percentages are calculated as the proportion of distinct codes or records in each year that are coded using each coding type. Read V2 codes are excluded from the figure to improve visual clarity, as they represent the majority of records.

0.651–0.656). Compared to quintile 1, the most deprived quintile, all other quintiles had adjusted odds ratios between 0.98 and 1.12.

For descriptive plots of mid-year population of Wales by SAIL-GP registration status from 1990 to 2024, stratified by sex, age, and WIMD see S1–S3 Figs in the Supporting information.

## Trends of GP activity

Fig 7 shows the average daily rates across five categories of GP activity per 100,000 people per month from January 2000 to December 2024 (derived from the WLGP RRDA), with an additional set of panels showing the trends over the COVID-19 pandemic. Fig 8 contains the estimated rate ratios comparing activity each year to 2019.

For consultations, we saw average daily rates rise from 18,560 per 100,000 people in January 2000–35,192 in December 2024 (+90%). During the COVID-19 pandemic, observed rates dropped below the expected rate for 12 months, reaching a low in April 2020 at 48.2% below expectation. Since the end of the COVID-19 pandemic, rates have recovered and have consistently been higher, we estimate the rate of consultations in 2024 to be +8% greater than that in 2019 (OR 1.076, 95% CI 1.073–1.079).

For GP events in which only a prescription was recorded, we saw a steady increase in the daily rates from 21,543–44,568 (+107%) between January 2000 and December 2024. During the COVID-19 pandemic, rates generally increased as well as fluctuated more than previously observed. Since the COVID-19 pandemic, rates have effectively returned towards 2019-levels (2024 OR 1.005, 95% CI 1.0051–1.0053).

Daily rate of vaccinations was always highest in October, and up until the COVID-19 pandemic had been slightly increasing over time, 9,680 in October 2000–13,759 in October 2019 (+42%). With the introduction of the COVID-19 vaccination programme, vaccination rates naturally massively exceeded the expected trend. Since winter 2022/23, when

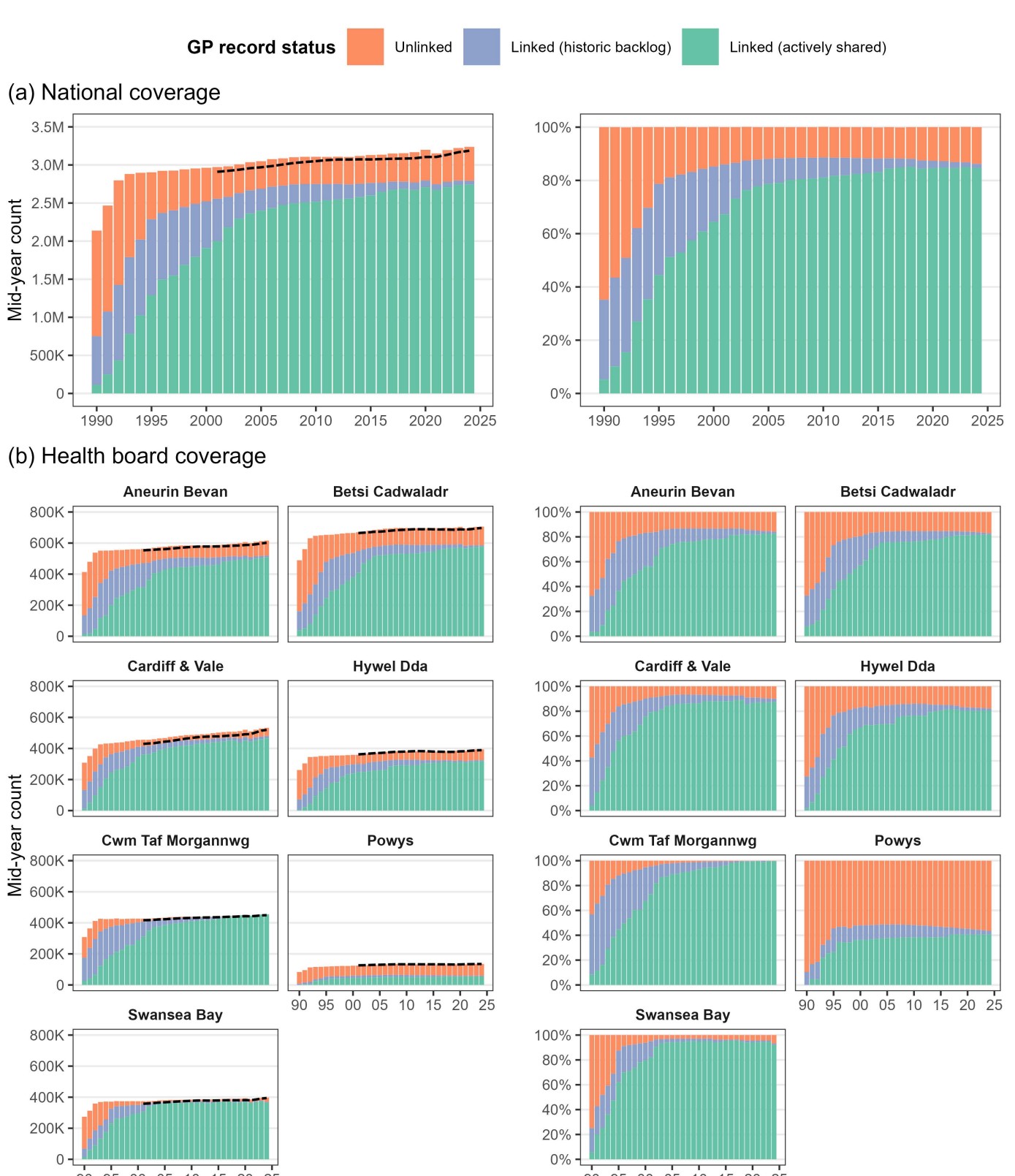

**Fig 5. Mid-year population of Wales by SAIL-GP registration status from 1990 to 2024, stratified by health board.** ONS mid-year population estimates overlayed as dashed lines from 1990 onwards for Wales, and from 2001 at the health board level.

**Table 3. Descriptive counts of all those living in Wales and the percentage of all those with GP record linkage for 1990, 2000, 2010, and 2020.**

| Total | | 1990 | 2000 | 2010 | 2020 | 2024 |
|---|---|---|---|---|---|---|
| | | 2,137,980 (35.2%) | 2,962,900 (85.1%) | 3,106,980 (88.5%) | 3,199,040 (87.4%) | 3,237,480 (86.2%) |
| **Sex** | Male | 1,056,340 (33.3%) | 1,465,740 (84.6%) | 1,552,580 (88.3%) | 1,598,170 (87.2%) | 1,616,100 (86.0%) |
| | Female | 1,081,650 (37.0%) | 1,497,160 (85.6%) | 1,554,400 (88.7%) | 1,600,880 (87.6%) | 1,621,380 (86.5%) |
| **Age** | 0 - 15 | 450,390 (37.7%) | 580,210 (88.2%) | 540,510 (89.7%) | 549,640 (88.2%) | 536,470 (87.1%) |
| | 16 - 34 | 491,490 (37.2%) | 736,080 (86.1%) | 754,510 (90.2%) | 737,040 (88.8%) | 731,260 (87.4%) |
| | 35 - 49 | 453,750 (37.6%) | 607,870 (86.2%) | 652,290 (88.7%) | 592,660 (87.7%) | 608,840 (86.5%) |
| | 50 - 64 | 385,480 (34.7%) | 534,630 (84.6%) | 600,340 (87.4%) | 653,920 (86.5%) | 663,370 (85.6%) |
| | 65 - 110 | 356,880 (26.8%) | 504,110 (79.6%) | 559,330 (86.1%) | 665,790 (85.9%) | 697,520 (84.8%) |
| **WIMD 2019** | Quintile 1 (most deprived) | 455,760 (36.6%) | 603,620 (86.8%) | 612,760 (91.1%) | 644,840 (90.4%) | 654,490 (89.2%) |
| | Quintile 2 | 442,430 (37.1%) | 592,870 (87.0%) | 614,220 (91.0%) | 629,930 (89.2%) | 635,650 (87.8%) |
| | Quintile 3 | 432,090 (33.9%) | 603,620 (83.9%) | 633,000 (87.4%) | 649,690 (86.4%) | 658,710 (85.3%) |
| | Quintile 4 | 422,990 (33.5%) | 588,200 (81.1%) | 625,590 (83.6%) | 642,140 (82.6%) | 649,430 (81.5%) |
| | Quintile 5 (least deprived) | 384,710 (34.6%) | 574,590 (87.0%) | 621,410 (89.6%) | 632,460 (88.4%) | 639,180 (87.5%) |
| **Health board** | Aneurin Bevan | 413,390 (32.6%) | 559,990 (83.2%) | 584,540 (86.7%) | 606,590 (85.5%) | 616,000 (84.2%) |
| | Betsi Cadwaladr | 489,100 (32.8%) | 664,980 (80.5%) | 696,840 (84.5%) | 704,090 (84.2%) | 705,900 (82.9%) |
| | Cardiff & Vale | 308,580 (42.8%) | 454,830 (90.2%) | 487,130 (93.2%) | 520,430 (91.0%) | 532,890 (90.1%) |
| | Cwm Taf Morgannwg | 308,430 (56.9%) | 426,720 (94.6%) | 439,710 (99.0%) | 453,110 (99.8%) | 455,560 (99.8%) |
| | Hywel Dda | 261,000 (27.5%) | 358,750 (83.1%) | 381,600 (86.0%) | 389,060 (83.1%) | 394,600 (82.2%) |
| | Powys | 83,390 (10.5%) | 123,970 (48.0%) | 133,000 (48.2%) | 133,130 (45.1%) | 135,070 (43.6%) |
| | Swansea Bay | 274,110 (25.2%) | 373,650 (93.7%) | 384,150 (97.1%) | 392,660 (95.7%) | 397,460 (93.2%) |

COVID-19 vaccinations were integrated with the winter flu program, rates have decreased compared to COVID-19 pandemic levels, but remained above pre-pandemic levels (2024 OR 1.75, 95%CI 1.63–1.89).

Similar patterns were observed for the two categories "patient review or monitoring" and "screening or assessment", in that rates were much smaller in scale but were severely disrupted by the pandemic. However, since the pandemic, rates have recovered to 2019 levels.

For a more detailed exploration of the seasonal and trend components of the different GP activities over time, as well as the outcome of the fitted splines, see S4–S19 Figs in the Supporting information.

## Discussion

This study describes the development of the WLGP RRDA, a curated and structured version of primary care data for Wales designed to improve analytic readiness, consistency, and scalability of research using WLGP data within the SAIL Databank. The RRDA enables more efficient use of routinely collected primary care data by applying methodical cleaning, code mapping, and normalisation techniques tailored to the Welsh healthcare context.

A major strength of this work is its large-scale, population-based scope. The primary care data comprises over 3.9 billion records between 1990 and 2024, covering more than three decades of general practice activity in Wales. This time span captures both pre and post-COVID-19 periods, enabling longitudinal research on trends in patient-practice interactions, clinical practices, clinical pathways, and healthcare utilisation.

We developed a comprehensive clinical code look-up that extends beyond the official Read V2 terminology to include additional coding systems (e.g., SNOMED-CT, local EMIS and Vision codes), as well as two supplementary categories: blank or invalid codes, and unknown codes. These additions improved the mapping of events to known descriptions,

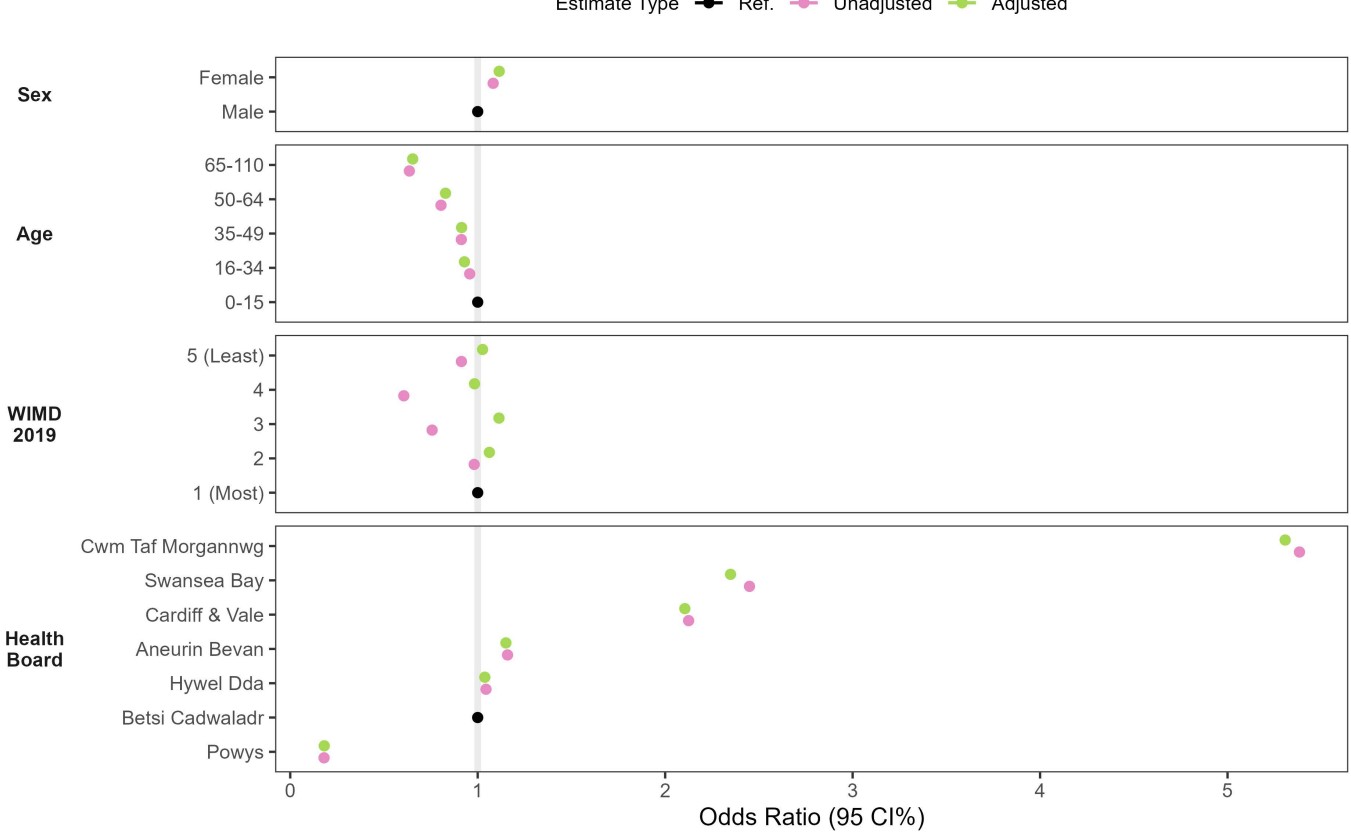

**Fig 6. Unadjusted and adjusted odds ratios of having linked GP records.** Adjusted estimates were obtained from multivariable models controlling for all other covariates under study.

allowing for more complete inclusion of recorded activity in analyses, especially in recent years when local and newer terminologies became more prevalent.

The RRDA's person-day-level structure supports faster querying and reduces memory load during analysis. While alternative strategies (e.g., splitting queries into smaller time windows) can sometimes be used with the original WLGP data, the RRDA simplifies this process and improves scalability for common research tasks.

A key contribution of this work is its reusability within the SAIL Databank. By sharing the code, metadata, and curated code lists used to generate the RRDA, we enable other researchers to adopt a consistent and transparent approach when working with the WLGP. While these methods are specific to the data and structure of SAIL, they offer transferable insights for curation efforts in other TREs and Secure Data Environments which hold similar data.

A key consideration when working with routine data is the extent of population coverage. Our evaluation of GP registration linkage over time showed that while linkage was incomplete in the early 1990s, it improved substantially by the 2000s and has remained stable since. As of 2024, over 86% of individuals recorded as living in Wales have linked primary care records, with variation by health board, age, and geography. Importantly, linkage rates were similar by sex and deprivation, though coverage remained lowest in the oldest age groups and in Powys, likely reflecting both population mobility and the distribution of non-data-sharing practices. Understanding these coverage patterns is essential for interpreting trends and identifying potential sources of bias, particularly in demographic or geographic subgroup analyses.

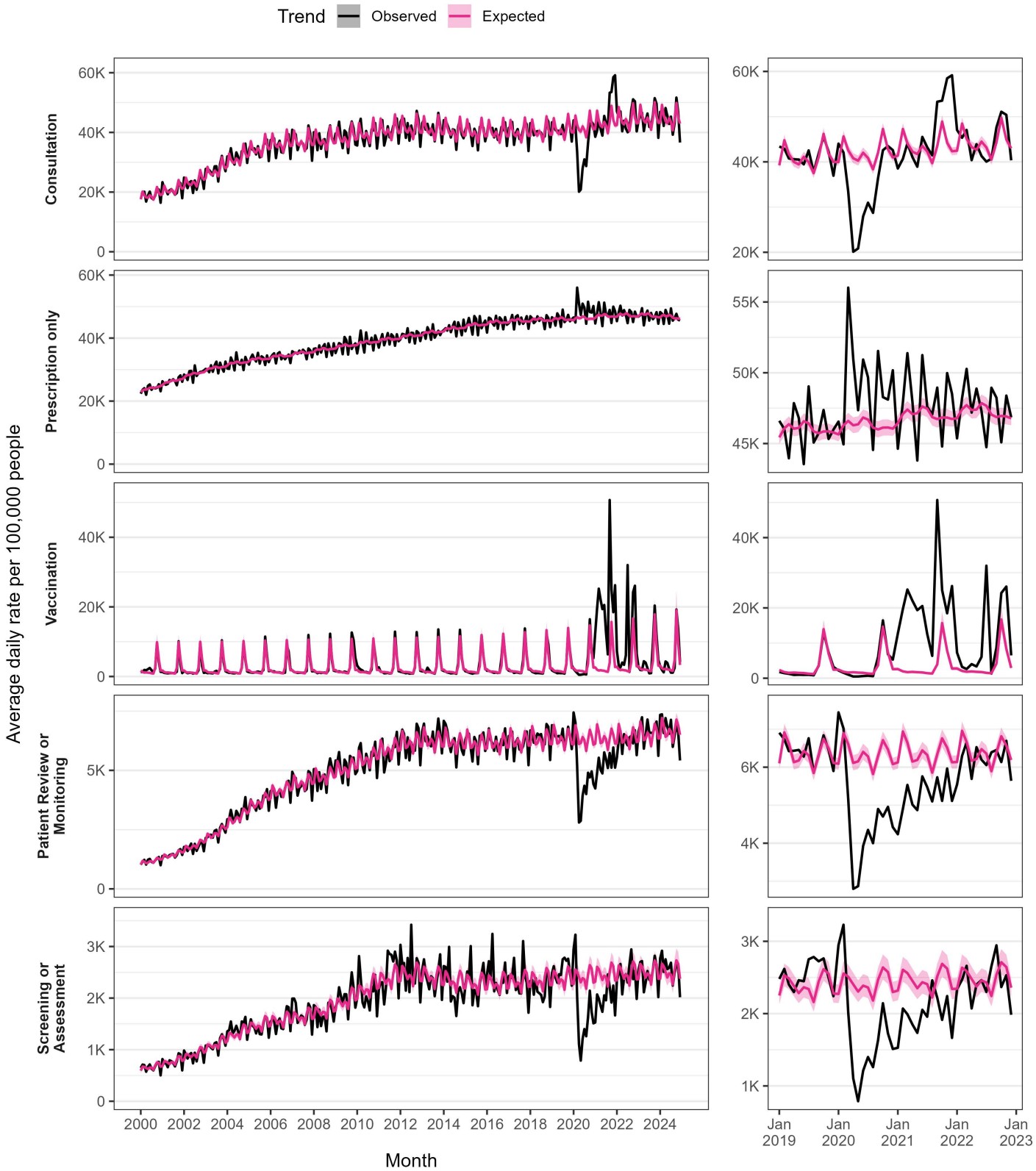

**Fig 7. Trends in daily GP activity per 100,000 people per month from January 2000 to December 2024.**

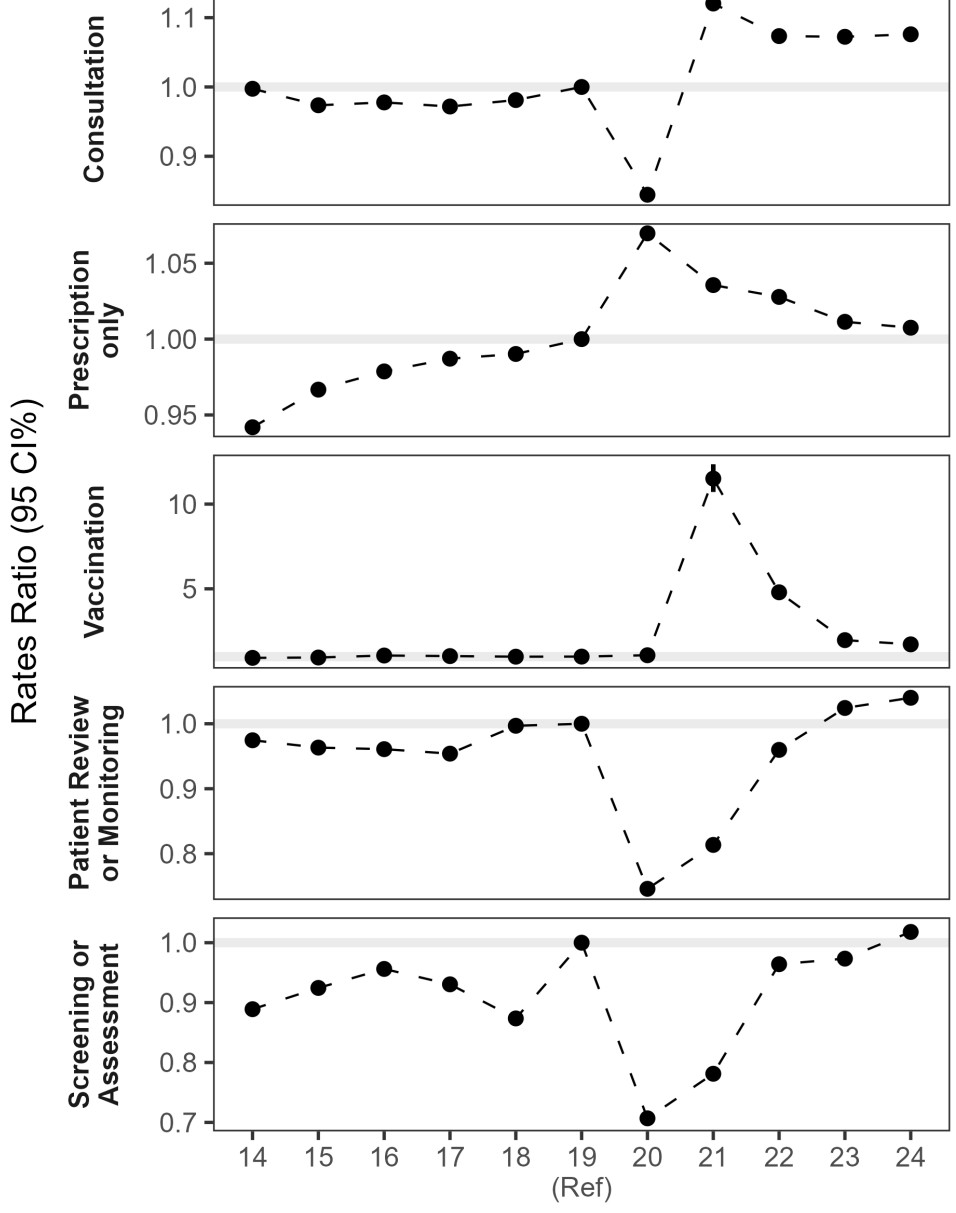

**Fig 8. Rate ratios of daily GP activities for years 2014 to 2024, relative to 2019.**

In addition to assessing coverage, we examined trends in general practice activity over the 25-year period encompassing the COVID-19 pandemic. Most activity types—including consultations and prescribing-only events, showed long-term growth, suggesting rising demand and workload in primary care. The COVID-19 pandemic caused sharp disruptions across all categories, with particularly steep declines during the early lockdown months. Vaccination rates, which typically peaked each October, rose sharply during the COVID-19 rollout and, though lower since, remain above pre-pandemic levels following their integration with flu campaigns. Activities such as "screening or assessment" and "patient review or monitoring" showed temporary reductions but have since returned to baseline levels. These patterns align with prior

evidence of disrupted chronic disease management and demonstrate the utility of longitudinal data assets like the RRDA for health system monitoring.

The study has some limitations. First, WLGP data does not cover every GP in Wales. Additionally, some clinical events remain unmapped due to unknown or poorly documented codes, particularly those from legacy systems, limiting the complete capture of recorded activity. The data also provides limited detail on the nature of each event, which complicates the classification of consultations. For example, some person-days contain only prescriptions or isolated test results, with no accompanying clinical or administrative entries, making it difficult to determine whether a consultation occurred. These have been excluded from the consultation category.

To analyse patterns of patient–practice interactions, we applied a structured four-layer approach to assign care provider, access mode, interaction type, and interaction details to each clinical event. However, because information on access mode and interaction type was often incomplete or inconsistently recorded, we focused our analysis on broader activity types—such as consultations, prescriptions-only events, and vaccinations—that could be more reliably derived. While this required some reliance on processing logic and metadata heuristics that are not externally validated, it enabled us to capture a more consistent picture of patient activity despite these data limitations.

Together, these findings highlight the value of a structured, reusable, and well-documented data asset for supporting timely and reliable research into primary care trends and outcomes in Wales. As with all individual-level data sources within the SAIL Databank, the WLGP RRDA includes the ALF, which enables secure linkage to other data sources such as secondary care, maternity and child health, mortality, census, and administrative data sources. This linkage capacity allows researchers to examine patient pathways and outcomes across the continuum of care. While our example focused on GP activity trends before and after the COVID-19 pandemic, the WLGP RRDA can be applied to a wide range of research questions where primary care data are relevant.

Building on these capabilities, the WLGP RRDA also enhances the usability of routine primary care data for research. Our evaluation provides insight into its strengths and limitations, particularly regarding data completeness and the potential impact of healthcare disruptions. When designing studies using the RRDA, researchers should carefully consider temporal coverage and population representativeness, as these factors may influence study findings. Although overall coverage is high, variation in which GP share data with SAIL could introduce selection bias. Two approaches can help address this. First, researchers can create inverse probability weights based on the likelihood of an individual having linked WLGP records. This would rebalance the study sample to better reflect the underlying population, as those with linked records but underrepresented characteristics receive greater weight in the analysis. Second, sensitivity analyses can be conducted by restricting studies to specific time periods or geographical regions with stable data coverage, then comparing these results with findings from the primary analytical approach to assess robustness.

## Supporting information

**S1 File. Illustrative examples of clinical code standardisation and duplicate resolution in the WLGP RRDA clinical code look-up.**
(DOCX)

**S1 Table. Descriptive counts and column percentages of all those with records of living in Wales, every 5 years from 1990 to 2024.**
(DOCX)

**S2 Table. Descriptive counts and column percentages of all those living in Wales with GP record linkage, every 5 years from 1990 to 2024.**
(DOCX)

**S3 Table. Percentages of all those living in Wales with GP record linkage, every 5 years from 1990 to 2024.**
(DOCX)

**S1 Fig. Mid-year population of Wales by SAIL-GP registration status from 1990 to 2024, stratified by sex.**
(TIF)

**S2 Fig. Mid-year population of Wales by SAIL-GP registration status from 1990 to 2024, stratified by age.**
(TIF)

**S3 Fig. Mid-year population of Wales by SAIL-GP registration status from 1990 to 2024, stratified by Welsh Index of Multiple Deprivation (WIMD 2019).**
(TIF)

**S4 Fig. STL decomposition of rates of averaged daily consultation between 2000 and 2024.**
(TIF)

**S5 Fig. Autocorrelation Function (ACF) of consultation rates.**
(TIF)

**S6 Fig. GAMM smoothing terms for calendar month and trend for consultation.**
(TIF)

**S7 Fig. STL decomposition of rates of averaged daily prescription-only events between 2000 and 2024.**
(TIF)

**S8Fig. Autocorrelation Function (ACF) of prescription-only event rates.**
(TIF)

**S9 Fig. GAMM smoothing terms for calendar month and trend for prescription-only events.**
(TIF)

**S10 Fig. STL decomposition of rates of average daily vaccinations per month between 2000 and 2024.**
(TIF)

**S11 Fig. Autocorrelation Function (ACF) of vaccination rates.**
(TIF)

**S12 Fig. GAMM smoothing terms for calendar month and trend for vaccination.**
(TIF)

**S13 Fig. STL decomposition of rates of average daily patient review or monitoring events per month between 2000 and 2024.**
(TIF)

**S14 Fig. Autocorrelation Function (ACF) of patient review or monitoring event rates.**
(TIF)

**S15 Fig. GAMM smoothing terms for calendar month and trend for patient review or monitoring events.**
(TIF)

**S16 Fig. STL decomposition of rates of average daily screening or assessment events per month between 2000 and 2024.**
(TIF)

**S17 Fig. Autocorrelation Function (ACF) of screening or assessment event rates.**
(TIF)

**S18 Fig. GAMM smoothing terms for calendar month and trend for screening or assessment events.**
(TIF)

**S19 Fig. Observed and expected trends across all GP activities between 2000 and 2024.**
(TIF)

## Acknowledgments

We would like to thank Lucy Robinson and Mattew Curds for their valuable input and support during the development of this work as part of the ADR Wales Major Societal Challenges research team. We also acknowledge all the data providers who make anonymized data available for research. This work uses data provided by patients and collected by the NHS as part of their care and support.

## Author contributions

**Conceptualization:** Hoda Abbasizanjani, Stuart Bedston, Ashley Akbari.

**Data curation:** Hoda Abbasizanjani, Stuart Bedston.

**Formal analysis:** Stuart Bedston, Hoda Abbasizanjani.

**Funding acquisition:** Ashley Akbari.

**Methodology:** Hoda Abbasizanjani, Stuart Bedston, Ashley Akbari.

**Project administration:** Hoda Abbasizanjani, Ashley Akbari.

**Supervision:** Ashley Akbari.

**Visualization:** Hoda Abbasizanjani, Stuart Bedston.

**Writing – original draft:** Hoda Abbasizanjani, Stuart Bedston.

**Writing – review & editing:** Hoda Abbasizanjani, Stuart Bedston, Ashley Akbari.

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
