## [Decision Letter · Decision Letter 0]

20 Oct 2025

Dear Dr. Abbasizanjani,

Thank you for submitting your manuscript to PLOS ONE. After careful consideration, we feel that it has merit but does not fully meet PLOS ONE’s publication criteria as it currently stands. Therefore, we invite you to submit a revised version of the manuscript that addresses the points raised during the review process.

We look forward to receiving your revised manuscript.

Kind regards,

Maria Christine Magnus, PhD

Academic Editor

PLOS ONE

Journal Requirements:

This work was supported by the ADR Wales programme of work. ADR Wales, part of the ADR UK investment, unites research expertise from Swansea University Medical School and WISERD (Wales Institute of Social and Economic Research and Data) at Cardiff University with analysts from Welsh Government. ADR UK is funded by the Economic and Social Research Council (ESRC), part of UK Research and Innovation. This research was supported by ESRC funding, including Administrative Data Research Wales (ES/W012227/1).

4. We note you have included a table to which you do not refer in the text of your manuscript. Please ensure that you refer to Table 2 and 3 in your text; if accepted, production will need this reference to link the reader to the Table.

Additional Editor Comments :

We have received feedback from two reviewers on your manuscript. They provide some useful suggestions on changes that should be made to improve the readibility and clarity of the manuscript. An important aspect that needs to be appropriately adressed is a specific description of how it is planned that this data based should be able to be accessed by other researchers; if this is the true ambition of the composition of the database. Please also make sure to correct errors in the references and the missing supplement information that reviewer 1 highlights.

Reviewers' comments:

Reviewer's Responses to Questions

**Comments to the Author**

1. Is the manuscript technically sound, and do the data support the conclusions?

Reviewer #1: Yes

Reviewer #2: Yes

2. Has the statistical analysis been performed appropriately and rigorously?

Reviewer #1: I Don't Know

Reviewer #2: Yes

3. Have the authors made all data underlying the findings in their manuscript fully available?

Reviewer #1: No

Reviewer #2: Yes

4. Is the manuscript presented in an intelligible fashion and written in standard English?

Reviewer #1: Yes

Reviewer #2: Yes

Reviewer #1: The paper describes a thorough and rigorous attempt to make a database that is usable for researching trends and healthcare utilisation in Primary Care. This is not something that is easy to do. I am not particularly familiar with GAM but as far as I can tell they appear to have been applied correctly.

Standard of English is good although there is a very extensive use of abbreviations, and these aren’t explained in Figure or table legends. The Results are confusing as they appear to contain some additional detail on the methods.

I have made some specific comments below.

For readability consider a supplementary table containing abbreviations. Consider not using abbreviations when there are few occurrences e.g. Lower layer Super Output Areas (LSOA) only occurs once .

P8 Step 1 to P10 Step 4. I find the use of “record” confusing. I think that the authors are referring to database records as to complete patient records, but this should be clarified.

P15 Lines 330 to 332. “WLGP coverage was defined as the number of individuals with shared records”. It’s not clear to me how this defines coverage. Records shared with what?

P18 Lines 385-388. Not sure why this text is here. It doesn’t seem like Results.

P18 Lines 390-396. Again I’m not sure that this is the right place for this paragraph. Much of it appears to be Method rather than result.

P26 Lines 540 to 544. It would be nice to see some more specific suggestions of how the database could be utilised, beyond the example of GP activity pre- and post-Covid.

Reviewer #2: This manuscript, 'Creating a Research-Ready Data Asset version of primary care data for Wales and investigating the impact of COVID-19 on utilisation of primary care services', presents a substantial piece of work. Some comments are as follows:

INTRODUCTION

There is some repetition around the description of online consultations. Lines 72-72 read “While these changes preserved access during lockdown, they also introduced questions about the long-term impact on patient care. And lines 76-77 read “While this preserved access during lockdowns, it also raised concerns about equity, diagnostic accuracy, and continuity of care.

Please clarify sentence at line 85: “At the same time, activity types in primary care also changed.” At the same time as what? The previous sentence mentions both before COVID-19 and during the pandemic.

In line 138 the authors state that “the motivation behind this is twofold” and then say firstly… secondly…. and thirdly. The first two points describe motivations, while the third (“thirdly…”) seems to refer to an action taken rather than a motivating factor.

METHODS

Lines 155-157 say: “The WLGP includes records for all patients registered with Welsh GPs, for the GPs who have agreed to share data with the SAIL Databank, as GPs are the data owners and must individually consent to contribute”. This link (https://research.senedd.wales/research-articles/patient-record-sharing-clearing-up-the-confusion/) however suggests individual patients may also be able to opt out: “Welsh patients can only opt-out by making a request to their GP”. Please clarify this in the text.

Line 163. Minor language/style suggestion: in scientific writing “data” is traditionally treated as a plural noun (i.e. “data are …”, “these data …”) especially in epidemiology / health data research. Suggest changing “data was” to “data were”. Please also check for use of “this data” in the rest of the manuscript (to change to “these data”).

Line 178. Suggest defining the abbreviation “SNOMED” at first use.

Line 206: The sentence describing the WDSD practice ID reads a bit awkwardly “(called WDSD practice ID in the paper)”. Consider rephrasing, although, looking through the manuscript, I can only find use of “practice ID” and not “WDSD practice ID”, so perhaps this detail is not needed?

Line 223: What is meant by “For any codes used in WLGP that did not fall into one of these categories (i.e., unverified or invalid codes), we created a new category and included them in the look-up”? Could you clarify what this new category was called?

Lines 228-229. The authors standardised the format of additional Vision codes to the 5-character level, either by removing extra characters or adding a "." to the right if the code was shorter than five characters. Could they clarify if the new standardized codes were unique i.e. is there a risk that two originally distinct codes could become identical after truncation? The whole of step 5 is slightly complicated. Consider providing some examples of how codes were standardised, truncated/padded, or prioritised, with descriptions, in the supplementary material.

Figure 1. Consider briefly clarifying in the figure legend or a note that the “foreign key” refers back to the primary key in another table (to help readers unfamiliar with database terminology).

Line 259: The paragraph in the Methods section beginning with “Understanding how patients interact with primary care services…” reads more like a rationale or background statement than a methodological description. As this is the Methods section, it would be clearer to focus on the procedural aspects of what was done.

Line 288: Please clarifying why SNOMED codes were excluded.

Lines 315-316: Where additional primary care person-day event categories were defined, are these mutually exclusive, or can a single event belong to multiple categories?

Line 323: It would be helpful to clarify what is meant by “failed encounters.” For example, could this refer to patients who did not attend scheduled appointments, cancelled appointments, or other types of incomplete interactions?

Line 334: It would be helpful to explicitly state the unit of analysis in the Generalised Additive Mixed Models. I note at the end of the paragraph it says a fuller exploratory analysis is provided in Section 1 of the Supplementary Material, but I could only find supporting tables and figures, not “Section 1”.

Line 336: Can you define AR(1) (e.g., “autoregressive of order 1”) when it is first introduced.

Line 347: Please also specify the unit of analysis for this analysis.

Lines 355-357: COVID-19 disruption periods are mentioned, but it might help to explain why vaccination exclusions are slightly different (2020–2021 vs 2021–2022).

RESULTS

Line 395: Please correct link (Error! Reference source not found).

Line 399. When referring to/describing the Figure in the text, please say whether you are referring to panel a) or b). The caption for Figure 4 could be clarified/expanded to specify what the percentages are calculated from. For example, I think the bars represents the % of unique clinical codes not in the official Read V2 code set / the % of total records using such codes.

Line 419-420: (Error! Reference source not found.)

Line 430: (Error! Reference source not found.)

Line 446: The figure title includes “(WIMD 2019)” but since the plot also includes other variables (age, sex, and health board), is (WIMD 2019) needed in the title or could it be removed? Could a note be included under the figure to say what has been adjusted for in the adjusted ORs?

Line 462-463: The text says, “For GP events in which only a prescription was recorded, we saw a steady increase in the daily rates from 21,543 to 44,568 (+207%) between January 2000 and December 2024”. Is the 207% correct? Should it be 107% increase?

Line 467-468: Similar to the comment above, the text says, “Daily rate of vaccinations was always highest in October, and up until the COVID-19 pandemic had been slightly increasing over time, 9,680 in October 2000 to 13,759 in October 2019 (+142%)”. Is 142% correct? Should it be 42% increase? Or it could be phrased 1.42 times higher?

DISCUSSION

Can the authors make recommendations for future researchers who may want to use this resource? For example, could the RRDA be linked with secondary care datasets to enable more comprehensive analyses of patient pathways and outcomes. Additionally, given the variation in data quality/completeness over time, how this should be considered when designing longitudinal studies using this resource?

**Do you want your identity to be public for this peer review?** For information about this choice, including consent withdrawal, please see our Privacy Policy

Reviewer #1: No

Reviewer #2: No

---

## [Author Response · Author response to Decision Letter 1]

30 Oct 2025

Journal Requirements:

RESPONSE: We confirm that the manuscript has been prepared in accordance with the PLOS ONE formatting and style requirements, and file naming conventions follow the journal’s guidelines.

RESPONSE: We confirm that our existing Data Availability statement already addresses these points. The data used in this study are held within the SAIL Databank, and access is subject to governance and approval by the independent Information Governance Review Panel (IGRP). As detailed in the statement, data cannot be publicly shared due to legal and ethical restrictions related to patient confidentiality, but researchers can request access through the SAIL application process (https://saildatabank.com/data/apply-to-work-with-the-data/). All relevant contact details, governance information, and guidance on data access are available on the SAIL Databank website (https://saildatabank.com).

This work was supported by the ADR Wales programme of work. ADR Wales, part of the ADR UK investment, unites research expertise from Swansea University Medical School and WISERD (Wales Institute of Social and Economic Research and Data) at Cardiff University with analysts from Welsh Government. ADR UK is funded by the Economic and Social Research Council (ESRC), part of UK Research and Innovation. This research was supported by ESRC funding, including Administrative Data Research Wales (ES/W012227/1).

RESPONSE: We confirm that the funders had no role in study design, data collection and analysis, decision to publish, or preparation of the manuscript. This statement has been added to the cover letter as requested.

4. We note you have included a table to which you do not refer in the text of your manuscript. Please ensure that you refer to Table 2 and 3 in your text; if accepted, production will need this reference to link the reader to the Table.

RESPONSE: We have reviewed the manuscript and ensured that all tables, including Tables 2 and 3, are appropriately referenced in the text.

Additional Editor Comments:

We have received feedback from two reviewers on your manuscript. They provide some useful suggestions on changes that should be made to improve the readibility and clarity of the manuscript. An important aspect that needs to be appropriately adressed is a specific description of how it is planned that this data based should be able to be accessed by other researchers; if this is the true ambition of the composition of the database. Please also make sure to correct errors in the references and the missing supplement information that reviewer 1 highlights.

RESPONSE: We confirm that all reviewer comments and editorial suggestions have been addressed, including corrections to references and the missing supplementary information.

Regarding data access, as detailed in the Data Availability statement, the data used in this study are held within the SAIL Databank and access is governed by the independent Information Governance Review Panel (IGRP). Due to legal and ethical restrictions related to patient confidentiality, data cannot be publicly shared, but researchers can request access through the SAIL application process (https://saildatabank.com/data/apply-to-work-with-the-data/). All relevant contact details, governance information, and guidance on data access are available on the SAIL Databank website (https://saildatabank.com).

We have also expanded the Discussion section to outline how future researchers can use or build upon the WLGP RRDA, including its potential for linkage with other datasets within SAIL to support broader health and care research.

Response to each point raised by reviewers have been uploaded as a seperate file.

---

## [Decision Letter · Decision Letter 1]

26 Nov 2025

Creating a Research-Ready Data Asset version of primary care data for Wales and investigating the impact of COVID-19 on utilisation of primary care services

PONE-D-25-51406R1

Dear Dr. Abbasizanjani,

We’re pleased to inform you that your manuscript has been judged scientifically suitable for publication and will be formally accepted for publication once it meets all outstanding technical requirements.

Kind regards,

Maria Christine Magnus, PhD

Academic Editor

PLOS ONE

Additional Editor Comments (optional):

Reviewers' comments:

Reviewer's Responses to Questions

**Comments to the Author**

Reviewer #1: All comments have been addressed

Reviewer #2: All comments have been addressed

2. Is the manuscript technically sound, and do the data support the conclusions?

Reviewer #1: Yes

Reviewer #2: (No Response)

3. Has the statistical analysis been performed appropriately and rigorously?

Reviewer #1: I Don't Know

Reviewer #2: (No Response)

4. Have the authors made all data underlying the findings in their manuscript fully available?

Reviewer #1: No

Reviewer #2: (No Response)

5. Is the manuscript presented in an intelligible fashion and written in standard English?

Reviewer #1: Yes

Reviewer #2: (No Response)

Reviewer #1: I have checked the revised manuscript and author responses, and the authors have adressed my comments adequately

Reviewer #2: (No Response)

**Do you want your identity to be public for this peer review?** For information about this choice, including consent withdrawal, please see our Privacy Policy

Reviewer #1: No

Reviewer #2: No

---

## [Editor Report · Acceptance letter]

1 Dec 2025

PONE-D-25-51406R1

PLOS ONE

Dear Dr. Abbasizanjani,

I'm pleased to inform you that your manuscript has been deemed suitable for publication in PLOS ONE. Congratulations! Your manuscript is now being handed over to our production team.

Kind regards,

on behalf of

Dr. Maria Christine Magnus

Academic Editor

PLOS ONE